# Ageing Evaluation of Foamed Polymer Modified Bitumen with Bio-Flux Additive

**DOI:** 10.3390/ma16062167

**Published:** 2023-03-08

**Authors:** Piotr Radziszewski, Adam Liphardt, Michał Sarnowski, Karol J. Kowalski, Piotr Pokorski, Katarzyna Konieczna, Jan B. Król, Marek Iwański, Anna Chomicz-Kowalska, Krzysztof Maciejewski, Mateusz M. Iwański, Maciej Michalec

**Affiliations:** 1Institute of Road and Bridges, Faculty of Civil Engineering, Warsaw University of Technology, 00-637 Warsaw, Poland; 2Department of Transportation Engineering, Faculty of Civil Engineering and Architecture, Kielce University of Technology, 25-314 Kielce, Poland; 3Department of Building Engineering Technologies and Organization, Faculty of Civil Engineering and Architecture, Kielce University of Technology, 25-314 Kielce, Poland; 4Zakład Robót Drogowych DUKT Sp. z o.o., 26-052 Nowiny, Poland

**Keywords:** foamed bitumen, fluxing additive, highly modified asphalt binder, ageing, DSR

## Abstract

This paper presents the results of an analysis of the changes in the stiffness of asphalt binders modified with a bio-flux additive and subjected to the processes of foaming and short-term ageing. The purpose of the analyses was to determine changes in the rheological properties of asphalt binder as a result of technological processes characteristic of hot and warm mix asphalt technology. Three asphalt binders with similar penetration but varying degrees of polymer modification were tested: 50/70, 45/80–55 polymer-modified bitumen, and 45/80–80 highly modified bitumen. Tests were carried out on four groups of binders: original binders, foamed binders after 14 days of storage, non-foamed binders after Rolling Thin Film Oven Test (RTFOT) ageing, and foamed binders after 14 days of storage subjected to RTFOT ageing. The master curves of the complex shear modulus G* were analysed, and three indexes of binder stiffening were determined, characterising the investigated effects. The tests showed that some of the stiffening indices significantly depended on the degree of polymer modification and the content of the bio-flux additive. Moreover, it was found that the foaming process in the case of paving-grade bitumen and polymer-modified bitumen did not contribute to the additional stiffening of the binders.

## 1. Introduction

Some of the most sought-after technological solutions in modern asphalt technology are those that enable decreasing production and paving temperatures of asphalt mixtures. Bitumen foaming is now one of the most important and widely used technologies used in the production of asphalt mixtures at reduced temperatures (Warm Mix Asphalt—WMA) [1], which also permits the simultaneous use of additives in the asphalt binder.

Research conducted at the Warsaw University of Technology showed that the addition of plant-origin fluxing additives (bio-flux additives) causes bitumen liquefaction [2], which is particularly important for improving the workability of asphalt mixtures during technological processes. Bio-flux additive is a product of the rapeseed methyl esters (RME), which are subjected to oxidation in the presence of a metal catalyst and organic peroxide [3]. As stated in the work of Piłat et al. [2], the oxypolymerisation reaction includes the application of organic acid salts of cobalt as a metal catalyst and cumene hydroperoxide as a polymerisation promoter.

Applying asphalt binder additives can be utilised to achieve different goals, which may be as simple as softening the asphalt binder, reducing the asphalt mix processing temperatures, rejuvenating the properties of aged asphalt binder [4], or even mitigating the effects of binder ageing [5]. Commonly investigated additives such as petroleum oils, bio-oils [6,7], bio-polymers [8], and vegetable oil-based [9] additives usually decrease the stiffness of the asphalt binder [10,11,12] and may significantly impact other properties of the binders, such as temperature performance [13], fatigue resistance [14,15], and large strain performance [11], and these effects can be affected by binder ageing.

Bitumens are subjected to high-temperature technological processes, and during long-term road operations, they experience ageing-related changes that lead to the hardening of the binders. As a result, the bitumen gradually changes its viscoelastic properties. Both short- and long-term ageing have a remarkable impact on the thermodynamic properties of asphalt, as stated by Xiao and Huang [16]. The analysis of the literature and the results of the authors’ research show that the sensitivity of bitumen to short-term ageing is diverse and depends on many factors, such as type of bitumen, bitumen production technology, type of modification, etc. [17,18,19]. Reducing the content of the modifier in the binder leads to an increase in the negative changes in viscoelastic properties as a result of technological ageing. At low polymer contents, a continuous phase of bitumen with dispersed polymer is observed. 

With a high polymer content in the bitumen, the continuous phase is formed by the polymer and the dispersed phase by the bitumen. In the first case, the properties of the binder modified in the ageing process are determined by the bitumen, and in the second—by the type of polymer. In the case of low polymer content, ageing proceeds faster due to the advantage of a significant hardening of the initial bitumen over the plasticising effect of low polymer content. With a higher polymer content, the stiffening of the bitumen by oxidation is more limited. In addition, it should be taken into account that, as a result of ageing, a partial, thermal decomposition of the polymer most likely occurs. Consequently, the stiffness of the binder decreases to a small extent, while the action of the polymer as a bitumen oxidation inhibitor limits the oxidation reaction and ageing proceeds slower [18,19]. 

Another method of lowering the bitumen viscosity and temperature of technological processes is bitumen foaming. Foaming of asphalt binder by injection of water (also known as mechanical water foaming) occurs by injecting a pressurised stream of water into a bulk of hot (typically above 150 °C) asphalt binder or mixing it with asphalt binder in a “foaming chamber”. As a result of foaming, the viscosity decreases, which allows lowering the mixing and compaction temperatures of the asphalt mixture. Water in contact with the hot binder vaporises instantly, producing water vapour and increasing its volume by orders of magnitude. This leads to the formation of bitumen foam, a system in which water vapour is dispersed in a continuous phase of the bituminous binder. Bitumen foam, due to the complex rheological characteristics of asphalt binder and the physical and chemical interactions in this system, possesses unique features. Bitumen foam produced in this way has increased capacity to coat aggregates at lower temperatures; it also increases the workability and compactability of the mixture, resulting in the possibility of reducing the processing temperatures of asphalt mixtures with foamed bitumen [20]. The capability to produce asphalt mixtures at lowered temperatures with foamed bitumen is typically explained by the effects of increased volume due to the water vapour and air encapsulated in the foam, which make it easier to mix with the aggregates due to decreased viscosity [21]. Another potential significant mechanism may include the observed decreased asphalt binder absorption in foam mixtures, which leaves more free asphalt binder in the mixture at the time of mixing and compaction [22], which finds its manifestation in a decreased optimum asphalt binder content found in these mixtures [23].

Multiple studies have investigated the effects that foaming could potentially have on the chemical structure of bitumen. Based on the Fourier-transform infrared spectroscopy (FTIR) test results obtained by Namutebi et al. [24], it was stated that the foaming process does not significantly change the chemical properties of different bituminous binders due to the short time of the bitumen’s exposure to water and air flow. These observations were also confirmed in the work of Maciejewski and Chomicz-Kowalska [25]. Similarly, Bairgi et al. [26], who investigated a performance-graded asphalt binder, found that after foaming it had a lower sulfoxide concentration than a non-foamed binder, which potentially contributed to its lower oxidative ageing potential or lower stiffness. Rheological characterisation of bitumens has also shown that the foaming procedure may cause changes in the high-temperature performance of bitumen binders; however, this effect was ascertained to be insignificant in the vast majority of cases [27,28]. 

Since bitumens can be modified with a variety of different additives prior to foaming and the foaming process can be applied by add-on modification at any asphalt plant, it can be easily and affordably combined with other techniques to enhance the characteristics of the binder, while some additives may also be used to enhance the foaming capabilities of the binders [29]. The application of bio-flux additive in combination with the technology of foamed bitumen and polymer-modified bitumen was described in the work of Iwański M. et al. [30].

To the knowledge of the authors, the impact of foaming technology with bio-flux addition on the rheological properties of aged binders (unmodified and highly modified with polymers) has not been sufficiently recognised so far. The objective of the research presented in this article was to determine the impact of short-term ageing and foaming on the rheological properties of unmodified and polymer-modified binders with the bio-flux additive and their changes over time. The binders were tested immediately after foaming and ageing, but also after an additional 14 days to evaluate the flux crosslinking process and the foamed bitumen recovery process. 

## 2. Materials and Methods

### 2.1. Bitumens and Bio-Flux Additive

Three petroleum bitumen binders were used in this study: 50/70 paving-grade bitumen, 45/80–55 polymer-modified bitumen, and 45/80–80 highly modified bitumen (HiMA). The binders were selected due to their similar penetration ranges and their differences in polymer modification levels. The characterisation of the base asphalt binders included in the present study is given in Table 1.

The aim of this selection of binders was to compare the influence of the polymer modification level on bitumens with similar penetration ranges.

The bio-flux additive was produced in laboratory conditions according to the patent number PL 214138 developed at the Warsaw University of Technology. Rapeseed methyl esters (RME) were used to produce the additive. For the purposes of the research, pure esters without anti-aging additives were obtained from the producer. 

To produce the bio-flux additive, the esters (RME) were oxidised using a special laboratory-scale reactor. The oxygenation time at 20 °C was 2 h. The air flow through the 1 kg oxidised RME sample was 500 L/h. Before starting the oxygenation process, fatty acid methyl esters were mixed with a cobalt catalyst and cumene hydroxide in accordance with the proportions shown in Table 2 [31].

### 2.2. Experimental Plan and Testing Methods

The experimental plan was set up to investigate the effects of short-term ageing, foaming, different levels of bitumen polymer modification, and different levels of fluxing additive modification. Tests were conducted in terms of the asphalt foaming and the time that elapsed after foaming:Asphalt binders not subjected to foaming (ORG),Asphalt binders not subjected to foaming after RTFOT ageing (RTFOT),Asphalt binders subjected to foaming, tested with a 14-day delay (F14d),Asphalt binders subjected to foaming and RTFOT ageing; tested with a 14-day delay (F14d_RTFOT).

The testing time delay of 14 days after foaming was derived from past experiences in the authors’ previous studies [2]. The selected period allowed the rheological properties of the asphalt binders to stabilise due to the recovery of the original properties of the foamed bitumen with the end of most of the flux-crosslinking processes. 

The investigated binders included 0%, 1%, 2%, and 3% of the bio-flux additive to evaluate how it affects their properties at different dose contents. The dosing rates were established based on preliminary testing that considered time-related stabilisation and experience from previous studies [31].

The rheological characterisation of asphalt binder in this study was conducted by means of an Anton Paar MCR101 dynamic shear rheometer. The complex shear modulus G* measurements were carried out in the 20–100 °C temperature range and 0.16–25 Hz frequency range. The strain level was set to maintain the linear viscoelasticity of measurements.

The RTFOT method, according to standard EN 12607-1, was used to prepare the aged binder samples.

The fluxing additive was mixed with the asphalt binders mechanically (if the binders were not foamed) or directly in the laboratory foamer device by the means of its overflow mechanism. The foaming device used in the study was the WLB-10S laboratory foamer (Wirtgen GmbH, Windhagen, Germany), and 2% foaming water content was used as in previous work [25], with air pressure set to 500 kPa and water pressure set to 600 kPa. The temperature of the asphalt binder before foaming was equal to 155 °C in all experiments, as per [25].

## 3. Results

### 3.1. Rheological Characteristics of the Complex Modulus G* of the Asphalt Binders

To assess the rheological properties of the original, aged, and foamed binders, the complex shear modulus G* was analysed. The results for this analysis were selected and compared for the 1.6 Hz frequency. Differences between binder types (polymer modification level) and bio-flux additive content were analysed.

The complex modulus G* of the 50/70 paving-grade bitumen is presented in Figure 1. There are three levels of bio-flux content for original binder (ORG), binder after RTFOT ageing (RTFOT), and foamed binder after 14 days of storage time (F14d), and their comparison is shown in Table 3.

The addition of the bio-flux additive significantly reduces the complex modulus G* of 50/70 paving-grade bitumen at all tested temperatures. At 20 °C, the reduction in G* modulus is over 40% with 1% bio-flux, over 65% with 2% bio-flux, and over 75% with 3% bio-flux. The process of foaming and storage for 14 days does not cause a significant change in the complex module G* compared to the original binder at all levels of bio-flux. This effect is caused by the fluxing effect of bio-flux stored with limited access to oxygen. 

RTFOT ageing causes the increase in the stiffness modulus as a function of the change in bio-flux content. At the test temperature of 20 °C, for 0% bio-flux, the increase in the stiffness modulus is about 80%; for 1% bio-flux additive, the increase is 140%; for 2% of bio-flux, it is over 200%; and for 3% of bio-flux, it is over 280%. This increase is due to the reaction of the 50/70 bitumen ageing oxidation process and the oxypolymerisation reaction of the bio-flux additive at high temperatures and with intensive oxygen access in the RTFOT ageing process.

The complex modulus G* of the 45/80–55 polymer-modified bitumen is presented in Figure 2 at three levels of bio-flux content: original binder (ORG), binder after RTFOT ageing (RTFOT), and foamed binder after 14 days of storage time (F14d) , and their comparison is shown in Table 4. 

The addition of the bio-flux additive causes a significant decrease in the complex modulus G* of 45/80–55 polymer-modified bitumen at all tested temperatures. At 20 °C, the reduction in modulus G* is almost 40% with 1% bio-flux, over 65% with 2% bio-flux, and over 80% with 3% bio-flux additive. The process of foaming and storage for 14 days does not cause a significant change in the G* complex module compared to the original binder at all bio-flux additive concentrations. This effect is caused by the fluxing effect of bio-flux stored with limited oxygen access, similar to 50/70 paving-grade bitumen. RTFOT ageing caused the increase in the stiffness modulus as a function of the change in bio-flux content. At 20°C temperature, for 0% bio-flux, this increase is about 85%; for 1% bio-flux, it is over 110%; for 2% bio-flux, it is over 180%; and for 3% bio-flux, it is over 250%. Ageing changes in polymer asphalt are smaller than in the paving-grade binder. They are most likely due to the partial thermal decomposition of a polymer, which reduces the stiffening effect of the binder and occurs as a result of age-related oxidation.

The complex modulus G* of the 45/80–80 highly modified bitumen is presented in Figure 3. on three levels of bio-flux content for original binder (ORG), binder after RTFOT ageing (RTFOT), and foamed binder after 14 days of storage time (F14d), and their comparison is shown in Table 5.

The addition of bio-flux reduces the complex modulus G* of 45/80–80 highly modified bitumen at all tested temperatures, but these changes are smaller than those in paving-grade bitumen and polymer-modified bitumen. At 20 °C, the reduction in G* modulus is over 35% with 1% bio-flux content, over 60% with 2% bio-flux content, and about 75% with 3% bio-flux content. That may be due to the morphology of highly modified bitumen, which is characterised by a continuous polymer phase that is less susceptible to liquefaction.

It should be emphasised that in the case of the 45/80–80 highly modified binder, changes in the stiffness modulus are noticeable at test temperatures above 40 °C, caused by the foaming process and storage for 14 days.

RTFOT ageing causes an increase in the stiffness modulus as a function of the change in bio-flux additive content. At 20 °C temperature, for 0% bio-flux, this increase is about 60%; for 1% bio-flux, it is over 110%; for 2% bio-flux, it is about 140%; and for 3% bio-flux, it is over 200%. It should be emphasised that in the case of highly modified bitumen, the increase in stiffness modulus is much smaller than in the case of polymer-modified bitumen and paving-grade bitumen. The aforementioned could be explained by the fact that, similarly to polymer-modified bitumen, ageing oxidation reactions of the modified binders and oxypolymerisation reactions of the bio-flux additive occur at high temperatures with intensive oxygen access in the RTFOT ageing process. Due to the high polymer content, there is less stiffening of the binder. As a result, the complex modulus G* increase due to short-term RTFOT ageing is the lowest in the case of 45/80–80 highly modified bitumen. 

### 3.2. Evaluation of the Master Curves of Tested Bitumen

Based on the results of the complex modulus G*, tested in a wide range of frequencies, the complex modulus master curves were determined for a 20 °C reference temperature using Excel software – version 2302. The master curves were determined using the temperature shift coefficient according to the Williams—Landel—Ferry (WLF) model, Equations (1) and (2), which was selected due to its adequacy in modelling asphalt binders [32,33,34]:(1)$${\log\alpha }_{T}=\frac{C_{I}\left(T-T_{\mathrm{ref}}\right)}{C_{\mathrm{II}}+T-T_{\mathrm{ref}}}$$
(2)$${\log\alpha }_{T}={\mathrm{logf}}_{\mathrm{red}}-\mathrm{logf}$$
where:α_T_—temperature shift coefficientf_red_—reduced frequencyf—actual test frequencyC_I_; C_II_—constant matching coefficientsT—test temperatureT_ref_—reference temperature

To determine the G* master curves, the sigmoidal function of the mechanistic-empirical pavement design guide was used. The sigmoidal function is described in Equation (3) [33,34].
(3)$$\log\left|G^{*}\right|=\delta +\frac{\alpha }{1+e^{\beta +\gamma \left({\mathrm{logf}}_{\mathrm{red}}\right)}}$$
where:G*—complex modulusδ—minimum value of G*α + δ—maximum value of G*β; γ—coefficients describing the shape of the sigmoidal functionf_ref_—reduced frequency

The complex modulus G* master curves of the analysed bitumen are presented in Figure 4, Figure 5 and Figure 6.

The fluxing properties of the bio-flux additive for paving-grade bitumen in the RTFOT ageing process are reduced but remain significant at a bio-flux content of 3% (Figure 4a,b). At the same time, for all levels of modification with the addition of bio-flux, an increase in the value of the complex modulus G* is observed, and these changes are greater with the increase in the content of the bio-flux addition in the binder. The higher rate of the bio-flux content increases the efficiency of the oxypolymerisation reaction. For the binder after foaming, the curve of changes in the complex modulus G* as a function of frequency (Figure 4c) is similar to the curve for 50/70 paving-grade bitumen before ageing (Figure 4a), which means that the foaming process does not affect the additional stiffening of the bitumen. It should be noted that changes in the complex modulus G* as a function of frequency for the foamed bitumen after RTFOT ageing (Figure 4d), especially in the lower frequency range, are greater than for the non-foamed binder (Figure 4b). It should be noted that in the case of the foamed bitumen, a significant fluxing effect can be observed when the bio-flux is added, especially at the 3% content.

The complex modulus G* master curves, of the 45/80–55 polymer-modified bitumen, are presented in Figure 5. 

The fluxing properties of the bio-flux additive to paving-grade bitumen in the RTFOT ageing process are reduced and remain significant only at a bio-flux content of 3% (Figure 5a,b). For the binder after foaming, the curve of changes in the complex modulus G* as a function of frequency (Figure 5c) differs from the curve for asphalt 45/80–55 before ageing in the range of the lowest and highest frequencies (Figure 5a). Slight differences occur in the mid-frequency range. The complex modulus values for the polymer-modified bitumen after foaming at the highest load frequencies are slightly lower than those of the non-foamed binder. It should be noted that the values of the complex modulus G* as a function of frequency for foamed bitumen after RTFOT ageing (Figure 4d) are lower at the highest analysed frequencies than for the non-foamed binder (Figure 4b). The effect of fluxing with the addition of bio-flux is comparable for non-foamed and foamed polymer-modified binder.

The complex modulus G* master curves of the 45/80–80 highly modified bitumen are presented in Figure 6. 

The fluxing properties of the bio-flux additive to paving-grade bitumen in the RTFOT ageing process are reduced and remain significant only at the bio-flux content of 3% (Figure 6a,b), similar to 45/80–55 polymer-modified asphalt. The increase in stiffness was found to be higher for binders with bio-flux additives than for binders without bio-flux additives. It should be noted that in the case of foamed bitumen subjected to RTFOT ageing, the fluxing effect shows for bio-flux contents of 2% and 3% in the range of higher frequencies, while at lower frequencies the fluxing effect disappears (Figure 6d). The results obtained for the foamed 45/80–80 bitumen after RTFOT ageing with 2% bio-flux content present lower G* complex modulus values than the 45/80–80 bitumen after RTFOT with 3% bio-flux addition. This phenomenon will be the subject of further research, e.g., chemical property assessment by FTIR.

In conclusion, it can be stated that the RTFOT ageing process, both in the case of foamed and non-foamed binders, increases the complex modulus G* value while reducing the fluxing effect. Moreover, it was noticed that the process of foaming and storing the binder for 14 days did not cause significant changes in the value of the complex modulus in the entire range of the analysed frequencies.

### 3.3. Quantitative Assessment of the Stiffening of Aged Bitumen

The stiffening indexes (SI) applied for the quantitative assessment of the changes in stiffness of the tested binders were defined as the ratios of G* values after RTFOT ageing ($G_{\mathrm{RTFOT}}^{*}$) or the foaming process without ($G_{F14d}^{*}$) and with RTFOT ageing ($G_{F14d\_\mathrm{RTFOT}}^{*}$) to the G* values for the corresponding original $(G_{\mathrm{ORG}}^{*}$) bitumen types. The SI values were calculated using Equations (4) and (5):(4)$$SI_{RTFOT}=\frac{G_{RTFOT}^{*}}{G_{ORG}^{*}}SI_{F14d}=\frac{G_{F14d}^{*}}{G_{ORG}^{*}}$$
(5)$$SI_{F14d\_RTFOT}=\frac{G_{F14d\_RTFOT}^{*}}{G_{ORG}^{*}}$$

For this calculation, the complex modulus G* was measured in the DSR rheometer in the 1.6 Hz frequency range and 20–100 °C temperature range. 

The stiffening indexes of the 50/70 paving-grade bitumen are presented in Figure 7. 

By analysing the stiffening indexes presented in Figure 7, it can be concluded that the RTFOT ageing process has the greatest impact on the stiffening of the 50/70 paving-grade bitumen, both in the case of non-foamed and foamed binders. The values of the stiffening indexes for bitumen only after foaming indicate that this process did not result in an additional increase in the complex modulus G*. The highest value of the stiffening indexes was obtained for bitumen with 3% of the bio-flux additive. With a decrease in the bio-flux content, a decrease in the stiffening indexes was observed. Both the non-foamed binder after RTFOT ageing and the foamed binder after RTFOT ageing were characterised by a similar course of stiffening index values as a function of temperature. Moreover, Figure 7a,c show that the stiffening index values are higher in the range of lower test temperatures (20–60 °C).

The stiffening indexes, of the 45/80–55 polymer-modified bitumen, are presented in Figure 8.

By analysing the stiffening indexes presented in Figure 8, it can be concluded that the RTFOT ageing process has the greatest impact on the stiffening of 45/80–55 polymer-modified bitumen, both in the case of non-foamed and foamed binders. Like paving-grade bitumen, the stiffening index values for foamed polymer-modified bitumen showed no increase in composite modulus G* due to foaming. Figure 8a,c show that the increase in stiffness indexes due to the RTFOT short-term ageing is greater the higher the content of bio-flux in the tested binder. In addition, a decrease in the stiffening index values is observed with the increase in the test temperature. The values of the stiffening indexes for the binder with 2% and 3% bio-flux additive content at the test temperature of 20 to 40 °C are lower for the foamed binder after RTFOT ageing than for the non-foamed binder after RTFOT ageing by about 0.5.

The stiffening indexes of the 45/80–80 highly modified bitumen are presented in Figure 9.

By analysing the stiffening indexes presented in Figure 9, it can be concluded that the stiffening of 45/80–80 highly modified bitumen, both in the case of non-foamed and foamed binders, is primarily caused by the RTFOT ageing process, in contrast to paving-grade bitumen and 45/80–55 polymer-modified bitumen. The increase in the stiffening indexes can also be observed for the binder with the addition of the bio-flux additive and foaming process. To explain this phenomenon, more detailed research on the chemical structure of the highly polymer-modified bitumens is needed.

Figure 9a,c show that the increase in stiffness indexes due to short-term ageing RTFOT is greater the higher the content of bio-flux in the tested binder. In addition, the decrease in stiffening indexes is observed with the increase in the test temperature, while for the foamed binder after ageing of RTFOT. The stiffening index remains almost the same in a temperature range from 20 °C to about 70 °C, particularly for the binder with 2% and 3% bio-flux.

It was found that the ageing index values decreased with increasing polymer content. The highest stiffening indexes were obtained for paving-grade bitumen, lower for 45/80–55 polymer-modified bitumen, and the lowest for 45/80–80 highly modified bitumen.

As a result of the partial thermal decomposition of polymers due to ageing, the stiffness of the binder is slightly reduced in modified and highly modified binders. At the same time, the action of the polymer as an asphalt oxidation inhibitor limits the oxidation reaction, and the ageing process is slower, as shown in the work of Radziszewski [17]. Therefore, asphalt stiffening is more limited with a higher polymer content.

Statistical analysis using Statgraphics 18 software was performed to verify if bitumen type, bio-flux content, and test temperature (independent variables) had a significant impact on the stiffening indexes (SI) of bitumens. The following dependent variables were defined: stiffening index of non-foamed bitumen after RTFOT ageing procedure (SI _RTFOT_), stiffening index of foamed bitumens (SI__F14d_), and stiffening index of foamed bitumens after RTFOT ageing procedure (SI__F14d_RTFOT_).

In the case of each stiffening index, multiple linear regression analysis was applied to describe the statistical relationship between independent and dependent variables. The 95% confidence interval was adopted for the statistical analysis. The probability *p*-values for the independent variables were determined as a result of the test statistic calculation to determine whether each factor (bitumen type, bio-flux content, test temperature) has a statistically significant influence on the value of the dependent variable. The effects of variables for which the *p*-values were less than or equal to the applied significance level α = 0.05 were considered statistically significant [35].

The results of the statistical analysis presented in Table 6 indicate the statistical significance of the influence of the bitumen type and bio-flux content on the SI_RTFOT_ and SI_F14d_RTFOT_ values (*p*-values equal to 0.000). Since the *p*-values corresponding to the variable of test temperature for SI_RTFOT_ and SI_F14d_RTFOT_ values are greater than 0.05, it can be concluded that the effect of the RTFOT ageing process on the changes of the G* complex shear modulus values of non-foamed and foamed binders is demonstrated regardless of the testing temperature. In the case of the stiffening index of foamed bitumens (SI_F14d_), the influence of bitumen type, bio-flux content, and test temperatures on the obtained values of the dependent variable was considered statistically insignificant (*p*-values equal to 0.870, 0.981, and 0.098, respectively). Therefore, it can be concluded that the foaming process does not significantly contribute to the increase in the stiffness of the binders.

## 4. Conclusions

This research investigated the influence of short-term ageing and foaming processes on changes in the properties of asphalt binders with different levels of polymer modification and the addition of a bio-flux additive. The research showed differences in the degree of ageing of the analysed binders with different types of polymer modification and changes in the content of the bio-flux additive. 

On the basis of the obtained results, it can be concluded:The bio-flux addition causes beneficial softening of the asphalt binders. The effectiveness of its action depends on the degree of polymer modification used in the mixture and the content of the bio-flux additive.The process of asphalt binder foaming does not cause significant changes in the complex module G* of paving bitumen and polymer-modified bitumen. Only for foamed, highly modified bitumen was an increase in the binder’s stiffness observed at high temperatures.The largest changes due to short-term ageing of RTFOT occur in 50/70 road bitumen, smaller in 45/80–55 polymer bitumen, and the smallest in 45/80–80 highly modified bitumen. As the polymer content in the asphalt binder increases, the binder’s resistance to short-term ageing increases.The significance of the influence of the type of binder, i.e., unmodified, modified, and highly polymer-modified bitumen, as well as the content of the bio-flux additive on the values of the determined ageing indicators, was determined.The degree of binder stiffening due to short-term ageing increases with increasing the bio-additive content. That is the efficiency result of the oxypolymerisation reaction with the increase in the content of the bio-flux additive.Depending on the temperature, the degree of binder stiffening varies. They are higher in the temperature range of 20–60 °C than in the temperature range of 60–100 °C.

## Figures and Tables

**Figure 1 materials-16-02167-f001:**
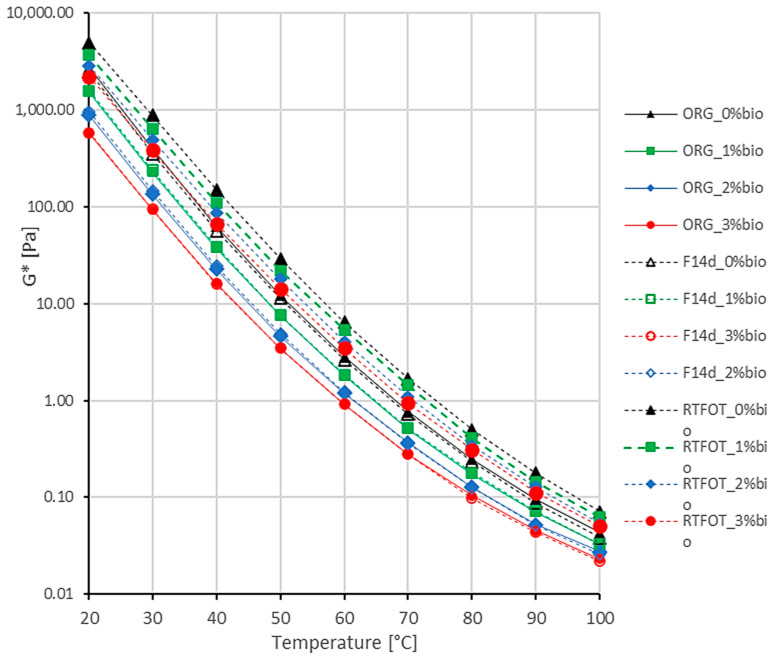
Complex modulus G* results of the 50/70 paving-grade bitumen.

**Figure 2 materials-16-02167-f002:**
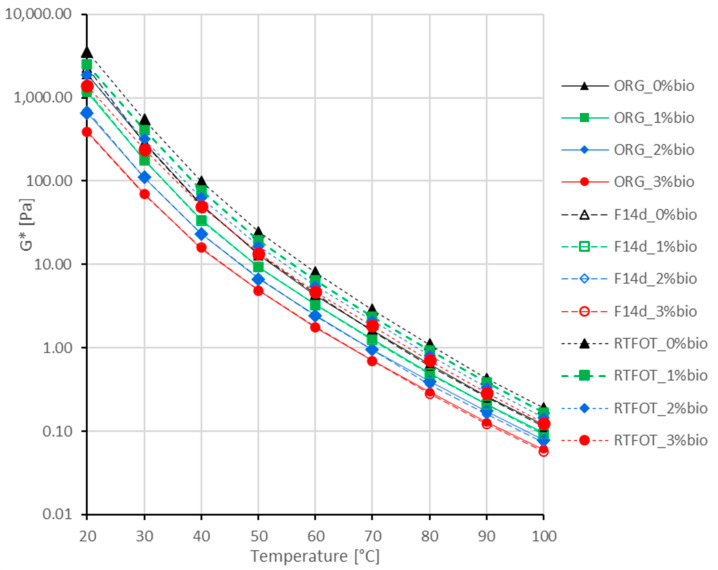
Complex modulus G* of the 45/80–55 polymer-modified bitumen.

**Figure 3 materials-16-02167-f003:**
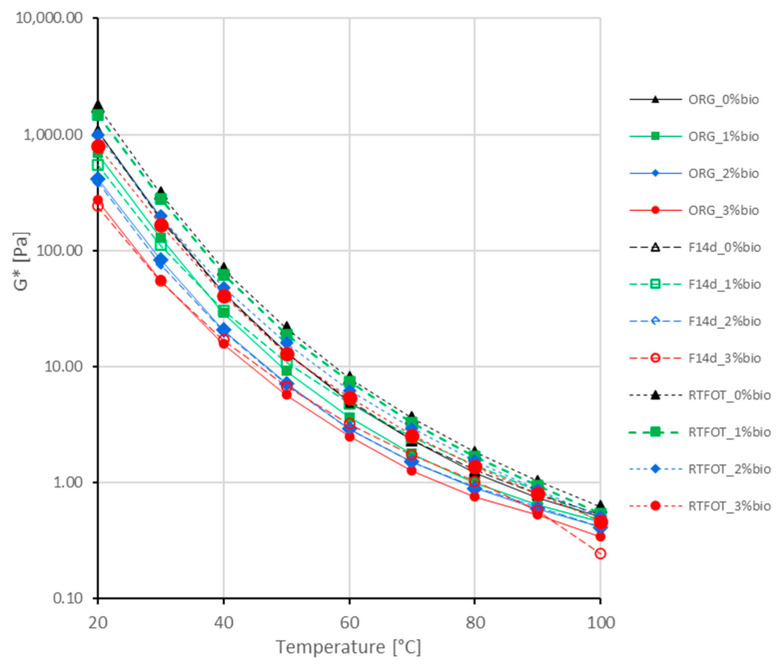
Complex modulus G* results from the 45/80–80 highly modified bitumen.

**Figure 4 materials-16-02167-f004:**
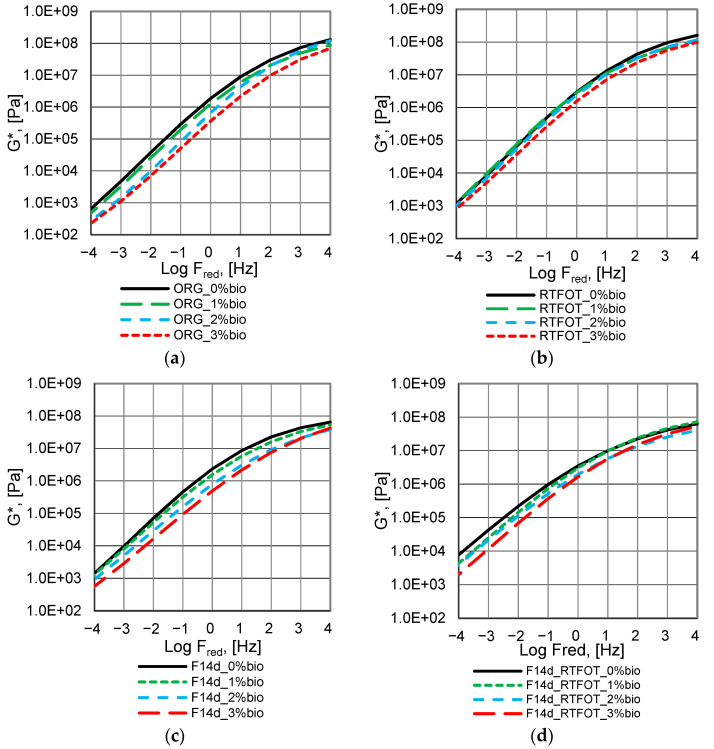
Master curves of complex modulus G* of the 50/70 paving-grade bitumen: (**a**) non-foamed original bitumen (ORG); (**b**) non-foamed after RTFOT ageing bitumen (RTFOT); (**c**) foamed original bitumen (F14d); (**d**) foamed after RTFOT ageing bitumen (F14d_RTFOT).

**Figure 5 materials-16-02167-f005:**
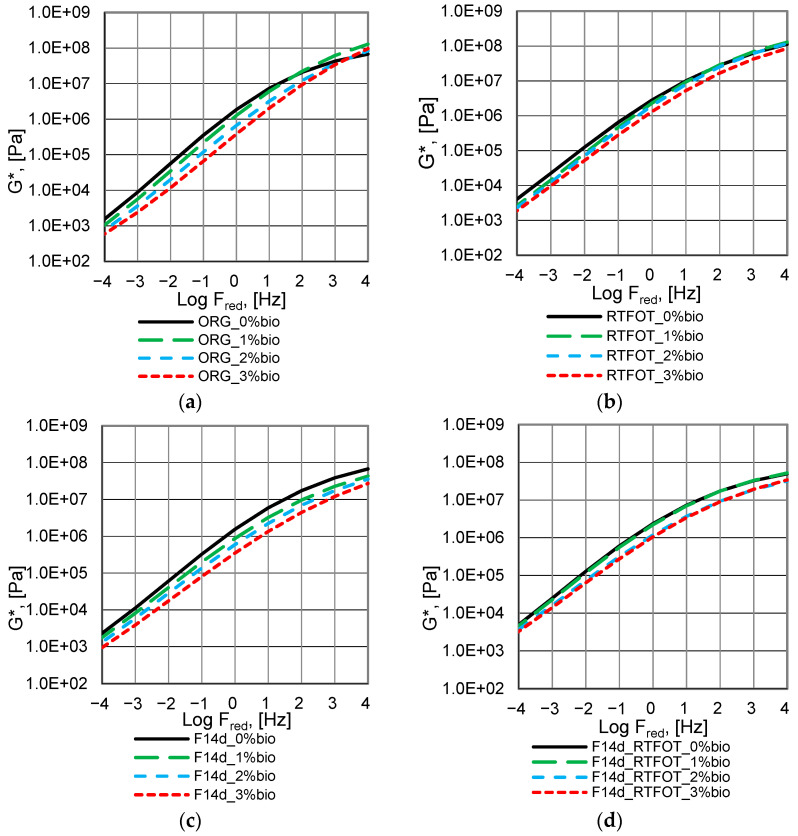
Master curves of complex modulus G* of the 45/80–55 polymer-modified bitumen: (**a**) non-foamed original bitumen (ORG), (**b**) non-foamed after RTFOT ageing bitumen (RTFOT), (**c**) foamed original bitumen (F14d), and (**d**) foamed after RTFOT ageing bitumen (F14d_RTFOT).

**Figure 6 materials-16-02167-f006:**
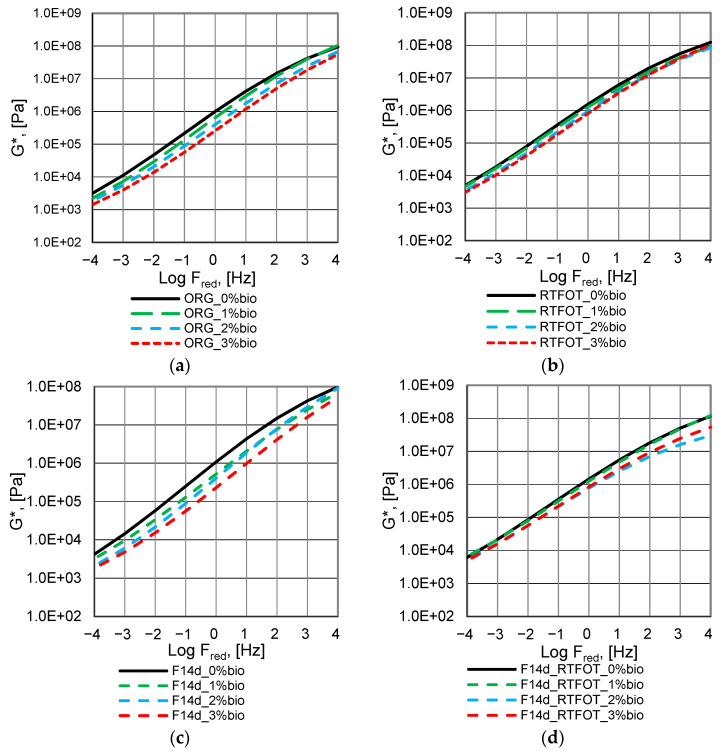
Master curves of complex modulus G* of the 45/80–80 highly modified bitumen: (**a**) non-foamed original bitumen; (**b**) non-foamed after RTFOT ageing bitumen; (**c**) foamed original bitumen; (**d**) foamed after RTFOT ageing bitumen.

**Figure 7 materials-16-02167-f007:**
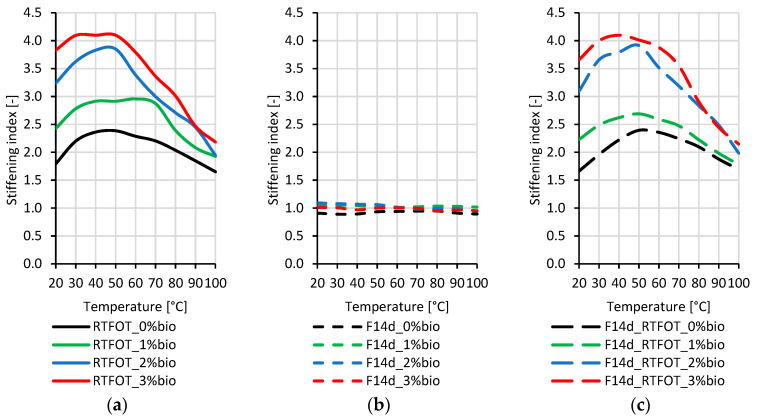
Stiffening index of the 50/70 paving-grade bitumen: (**a**) non-foamed after RTFOT ageing bitumen (RTFOT); (**b**) foamed original bitumen (F14d); (**c**) foamed after RTFOT ageing bitumen (F14d_RTFOT).

**Figure 8 materials-16-02167-f008:**
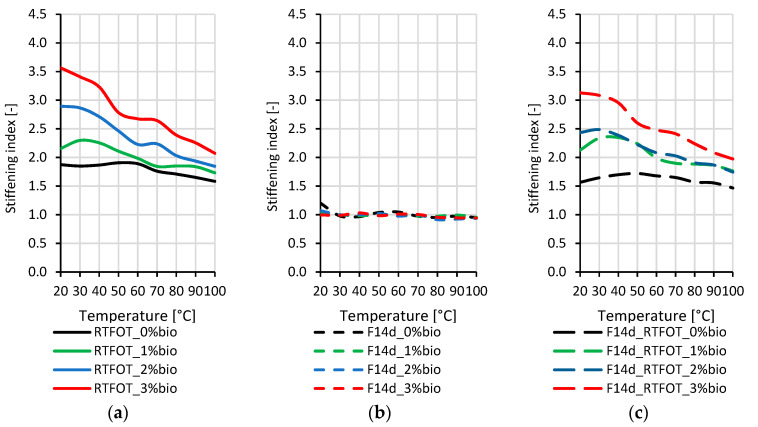
Stiffening index of the 45/80–55 polymer-modified bitumen: (**a**) non-foamed after RTFOT ageing bitumen (RTFOT); (**b**) foamed original bitumen (F14d); (**c**) foamed after RTFOT ageing bitumen (F14d_RTFOT).

**Figure 9 materials-16-02167-f009:**
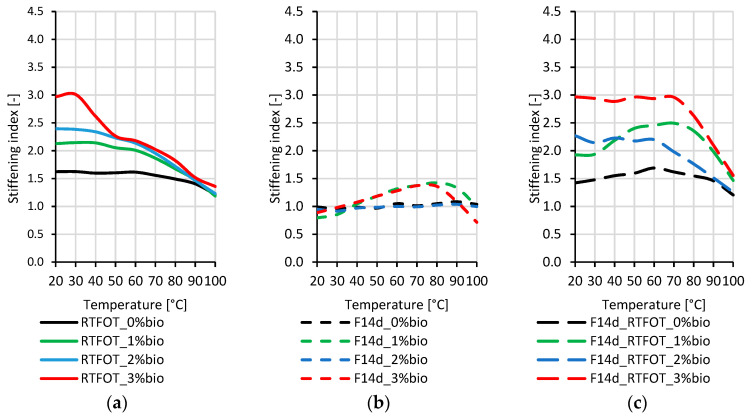
Stiffening index of the 45/80–80 highly modified bitumen: (**a**) non-foamed after RTFOT ageing bitumen (RTFOT); (**b**) foamed original bitumen (F14d); (**c**) foamed after RTFOT ageing bitumen (F14d_RTFOT).

**Table 1 materials-16-02167-t001:** Properties of the base bitumens.

Property	Unit of Measurement	Base Bitumen			Testing Method
		50/70	45/80–55	45/80–80	
Penetration at 25 °C	0.1 mm	65	71	75	EN 1426
Softening point by R and B	°C	48.2	57.8	95.5	EN 1427
Elastic recovery after RTFOT	%	-	83	92	EN 13398
High critical temperature:					EN 14770, EN 12607-1
Before RTFOT (G*/sin(δ) = 1.0 kPa)	°C	68.0	75.8	91.8	
After RTFOT (G*/sin(δ) = 2.2 kPa)	°C	68.1	73.5	81.3	
Low critical temperature:					EN 14771,
S = 300 MPa	°C	−16.8	−18.6	−22.2	EN 12607-1
M = 0.3	°C	−12.3	−15.3	−18.3	EN 14769

**Table 2 materials-16-02167-t002:** Proportions of additives used in the RME oxidation process.

Ingredient	Percentage Content (% m/m)
RME	98.9
Cobalt acetate tetrahydrate C_4_H_6_CoO_4_·4H_2_O	0.1
Cumene hydrogen peroxide C_6_H_5_CMe_2_OOH	1.0

**Table 3 materials-16-02167-t003:** Comparison of G* modulus results in 20 °C of the 50/70 paving-grade bitumen.

	Bio-Flux Content (%)	Test Temperature								
		20	30	40	50	60	70	80	90	100
Complex modulus of original binder (ORG) (kPa)	0	2760	401	63.5	12.2	2.8	0.77	0.25	0.096	0.043
	1	1540	228	37.4	7.55	1.8	0.51	0.17	0.070	0.032
	2	886	136.0	22.8	4.65	1.20	0.367	0.127	0.052	0.027
	3	577	93.8	16.4	3.49	0.918	0.280	0.104	0.045	0.023
Percentage stiffness decrease by bio-flux (%)	0	-	-	-	-	-	-	-	-	-
	1	44	43	41	38	35	34	30	27	25
	2	68	66	64	62	58	52	49	46	37
	3	79	77	74	71	68	63	58	53	47
Complex modulus of binder after RTFOT ageing (RTFOT) (kPa)	0	4950	882	150	29	6.5	1.7	0.50	0.18	0.071
	1	3740	634	109	22	5.4	1.5	0.42	0.15	0.062
	2	2870	493	87.3	17.9	4.1	1.1	0.34	0.13	0.053
	3	2210	384	67.2	14.3	3.5	0.9	0.31	0.11	0.050
Percentage Stiffness increase by RTFOT (%)	0	79	120	136	139	129	120	104	85	65
	1	143	178	191	191	196	187	139	108	93
	2	224	263	283	285	238	200	171	145	95
	3	283	309	310	310	279	236	201	147	118

**Table 4 materials-16-02167-t004:** Comparison of G* modulus results at 20 °C for the 45/80–55 polymer-modified bitumen.

	Bio-Flux Content (%)	Test Temperature								
		20	30	40	50	60	70	80	90	100
Complex modulus of original binder (ORG) (kPa)	0	1910	294	52.9	12.9	4.23	1.63	0.64	0.261	0.120
	1	1160	178	34.5	9.29	3.29	1.28	0.50	0.211	0.096
	2	653	111	23.3	6.70	2.46	0.947	0.400	0.174	0.079
	3	390	70.7	15.5	4.92	1.75	0.696	0.297	0.129	0.061
Percentage stiffness decrease by bio-flux (%)	0	-	-	-	-	-	-	-	-	-
	1	39	39	35	28	22	21	22	19	20
	2	66	62	56	48	42	42	38	33	34
	3	80	76	71	62	59	57	54	51	49
Complex modulus of binder after RTFOT ageing (RTFOT) (kPa)	0	3580	544	98.8	24.6	8.00	2.87	1.10	0.431	0.190
	1	2500	409	77.8	19.6	6.53	2.36	0.929	0.388	0.167
	2	1890	318	63.2	16.5	5.48	2.12	0.813	0.337	0.146
	3	1390	241	50.1	13.7	4.68	1.84	0.710	0.291	0.126
Percentage stiffness increase by RTFOT (%)	0	87	85	87	91	89	76	71	65	58
	1	116	130	126	111	98	84	85	84	73
	2	189	186	171	146	123	124	103	94	85
	3	256	241	223	178	167	164	139	126	107

**Table 5 materials-16-02167-t005:** Comparison of G* modulus results in 20°C of the 45/80–80 highly modified bitumen.

	Bio-Flux Content (%)	Test Temperature								
		20	30	40	50	60	70	80	90	100
Complex modulus of original binder (ORG) (kPa)	0	1090	192	43.5	13.4	4.94	2.31	1.22	0.733	0.510
	1	691	130	29.1	9.20	3.67	1.78	1.00	0.640	0.457
	2	415.0	83.5	20.6	7.17	2.92	1.51	0.896	0.595	0.412
	3	271.0	55.8	15.7	5.70	2.50	1.26	0.757	0.529	0.343
Percentage stiffness decrease by bio-flux (%)	0	-	-	-	-	-	-	-	-	-
	1	37	32	33	31	26	23	18	13	10
	2	62	57	53	46	41	35	27	19	19
	3	75	71	64	57	49	45	38	28	33
Complex modulus of binder after RTFOT ageing (RTFOT) (kPa)	0	1770	312	69.5	21.5	7.98	3.60	1.82	1.03	0.620
	1	1470	279	62.3	18.9	7.37	3.32	1.67	0.944	0.542
	2	994	199	48.2	16.0	6.24	2.95	1.54	0.872	0.507
	3	805	168	41.1	12.9	5.45	2.55	1.38	0.803	0.466
Percentage stiffness increase by RTFOT (%)	0	62	63	60	60	62	56	49	41	22
	1	113	115	114	105	101	87	68	48	19
	2	140	138	134	123	114	95	72	47	23
	3	197	201	162	126	118	102	82	52	36

**Table 6 materials-16-02167-t006:** Determined *p*-values for the variables.

Variable	*p*-Value		
	SI_RTFOT_	SI_F14d_	SI_F14d-RTFOT_
Bitumen type	0.000	0.870	0.000
Bio-flux content	0.000	0.981	0.000
Test temperature	0.073	0.098	0.841

## Data Availability

Data are available on request from the corresponding author.

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
