# Peer review of "Ageing Evaluation of Foamed Polymer Modified Bitumen with Bio-Flux Additive"

_materials, 2023, doi:10.3390/ma16062167_

Round 1
Reviewer 1 Report (Previous Reviewer 2)
the reviewer is satisfied to the improvement in the revised manuscript.
Author Response
Dear Reviewer,
We would like to voice our gratitude for the time and effort spent revising our paper titled “Ageing evaluation of foamed polymer modified bitumen with bio-flux additive”. We truly feel that your remarks are a significant contribution to the overall quality of our paper and enabled us to rectify its shortcomings.
After a thorough revision, we present you the corrected version of the manuscript for its assessment. Please find the detailed responses to your comments below.
Best regards,
Authors
Reviewer 1:
the reviewer is satisfied to the improvement in the revised manuscript.
Thank you for the generous comment.
Reviewer 2 Report (Previous Reviewer 3)
Manuscript is much better now, please make corrections requested in Word file and it will be acceptable.

Author Response
Dear Reviewer,
We would like to voice our gratitude for the time and effort spent revising our paper titled “Ageing evaluation of foamed polymer modified bitumen with bio-flux additive”. We truly feel that your remarks are a significant contribution to the overall quality of our paper and enabled us to rectify its shortcomings.
After a thorough revision, we present you the corrected version of the manuscript for its assessment. Please find the detailed responses to your comments below.
Best regards,
Authors
Reviewer 2:
Manuscript is much better now, please make corrections requested in Word file and it will be acceptable.
Comment [A1]: Please make Abstract to fit 200 words as required by Instructions for authors.
Thank you for the comment, we have modified the abstract to 200 words.
Comment [A2]: In first sentence you were mentioning asphalt modifications that ENABLE production at lower temperatures, but in second sentence you are getting to introduce technique that is PERFORMING at lower temperatures. This is not the same, and it is not related.
Thank you for the comment, we have modified the sentences.
Comment [A3]: This paragraph could end here.
Thank you for the comment, we have modified the paragraph.
Comment [A4]: “typically above 150oC” – you are mentioning “lowering the temperature” so please reformulate this sentence to sound like lowering the temperature.
Thank you for the comment, we have added the one more sentence to show clearly the lowering temperature benefits
Comment [A5]: “to coat aggregates of ___?____ at lower temperatures”
Thank you for the remark. We used ‘aggregates’ in the meaning of ‘building stones, small stones used in building’ as after Cambridge Dictionary (https://dictionary.cambridge.org/dictionary/english/aggregate)
Comment [A6]: “the foamed bitumen binder PG 64-22 contained…” . Since this is not the binder used in this work please find work with binder that you used and give reference to keep this sentence and it’s construction as a part of a work.
Thank you for the accurate remark. We have reformulated the sentence to more adequately convey the intended thought:
“Similarly, Bairgi et al. [26], who investigated a performance graded asphalt binder have found that after foaming it had lower sulfoxide concentration than a non-foamed binder, which potentially contributed to its lower oxidative aging potential or lower stiffness.”
Comment [A7]: In work cited 26 temperature was 100o and 135o, but above you wrote 150o.
Thank you for the accurate remark. We have corrected the citation to [27, 28].
Comment [A8]: Please give reference for reaction performed for bio-flux additive production.
Thank you for the comment, we have added the following reference: Król, J.B.; Niczke, Ł.; Kowalski, K.J. Towards understanding polymerisation process in bitumen bio-fluxes. Materials, 2017, 10, 9, doi: 10.3390/ma10091058.
Comment [A9]: Please add this as an aim of the work in last paragraph in section Introduction.
Thank you for the comment. We have supplemented the introduction as suggested.
Comment [A10]: It was expected that mixer was used for mixing binder and additive.
Thank you for the comment. The laboratory foamer used in the investigations when filled with the asphalt binder constantly pumps the binder through its system mixing and homogenizing at the same the asphalt binder contained in its reservoir. During the years of testing we have assessed that this mechanism is sufficient to mix and homogenize the asphalt binder with liquid additives and solid additives characterized by melting point significantly lower than the temperature of the asphalt binder (e.g. Sasobit).
Comment [A11]: This look more clear. My suggestion is to rotate table so on the far left column one below other would be: Bio flux content/ Complex modulus of original binder/ Percentage stiffness decreased by bio-flux…. And in the vertical columns put all results for all temperatures. That like all results will be visible for readers and chances to get citations are bigger. Please make table like that for all 3 bitumen samples.
Thank you for the comment. We have changed all three tables according suggestion
Comment [A12]: New paragraph.
Thank you for the comment.
Comment [A13]: Please carefully check every single reference once again and format it according to the instructions for authors given for journal Materials.
Thank you for the comment. We have checked all references and improved it according to instructions
Comment [A14]: Please add Country and pages to reference 17.
Thank you for the comment. We have added Country and pages to reference 17
Comment [A15]: Please fill the reference for conference as it is given in the Instructions for authors/References/Example 7 Conference Proceedings. Country and date is missing…
Thank you for the comment. We have added Country and date to reference 21
Comment [A16]:
No comment
Comment [A17]: Please cite reference 27 according to instructions for authors given for example 7 Conference Proceedings.
Thank you for the comment. We have changed reference 27 according to instructions for authors
Reviewer 3 Report (Previous Reviewer 1)
This paper has been revised based on the comments and should be ready for publication.
Author Response
Dear Reviewer,
We would like to voice our gratitude for the time and effort spent revising our paper titled “Ageing evaluation of foamed polymer modified bitumen with bio-flux additive”. We truly feel that your remarks are a significant contribution to the overall quality of our paper and enabled us to rectify its shortcomings.
After a thorough revision, we present you the corrected version of the manuscript for its assessment. Please find the detailed responses to your comments below.
Best regards,
Authors
Reviewer 3:
This paper has been revised based on the comments and should be ready for publication.
Thank you for the generous comment.
This manuscript is a resubmission of an earlier submission. The following is a list of the peer review reports and author responses from that submission.
Round 1
Reviewer 1 Report
In this paper, the aging evaluation of foamed polymer modified bitumen with bio-flux additive was investigated. Overall speaking, this is a very interesting paper. However, there are some problems which the authors should look into.
1. The authors have to look into the obvious grammar errors throughout the paper. For example, #Line 42 "can refreshing/changing".
2. #Line 107: The objective of this paper should be placed at the end of the introduction. The structure of introduction should be clear.
3. The knowledge gap of this paper is not well explained. The innovation of the paper should be emphasized.
4. The influence of aging on asphalt pavements should be reviewed with more details. For example, the long-term aging of asphalt will accelerate the moisture damage of asphalt pavements. ("Moisture damage mechanism and thermodynamic properties of hot-mix asphalt under aging conditions. ACS Sustainable Chemistry & Engineering, 10(45), pp.14865-14887.")
5. In the section of 2.2, you have a list of experiments. Have you reported all the results?
6. More indepth discussion on the mechanisms should be provided instead of only describing the experimental findings.
Reviewer 2 Report
The manuscript investigated the influence of bio-flux additives and foaming processes on changes in the rheology properties of asphalt binders. The authors did good research work and the results are very useful to guide the application of bio-flux additives and foamed asphalt in pavement engineering. The reviewer list the following comments to improve the manuscript:
(1) The abstract, the background is recommended to be added in the beginning;
(2) The full name needs to be shown when the abbreviations show for the first time, for example, RTFOT, DSR, etc;
(3) Some typos are found, for example, line 37, fluxing additive (additive), line 57, bio-flux, line 151, 2÷4%, lines 153&156, RTFO, etc
(4) The introduction is too long and needs to be shortened and focused on the topic of the manuscript;
(5) In Figures 4, and 5&6, what is the meaning of the arrows?
(6) The newly defined stiffening index is recommended to be defined clearly using the formula, and the meaning of the stiffening index also needs to be clarified;
(7) Line 522, what is p-value, it needs to be clarified.
Reviewer 3 Report
You can find revised manuscript as word document.
Please use references that are more impactful in scientific community, there is too much self citations and that leaves the impression that self citation is the aim of this work, and present your results as other authors do. Considering the work cited [5] this one looks like example of "salami publication".
